# Factor Structure, Validity and Reliability of the Intolerance of Uncertainty Scale -12 (IUS-12) in a Greek Undergraduate Sample

Gregoris Simos * and Anna Nisyraiou *

Department of Educational and Social Policy, School of Social Sciences, Humanities and Arts, University of Macedonia, GR-546 36 Thessaloniki, Greece
* Correspondence: gsimos@uom.edu.gr (G.S.); anisiraiou@uom.edu.gr (A.N.)

**Abstract:** Intolerance of Uncertainty (IU) is described as the tendency to avoid uncertain states and exhibit negative responses to uncertain situations on cognitive, emotional, and behavioral levels. It is considered a transdiagnostic cognitive bias that plays a role in developing and maintaining psychopathology. The Intolerance of Uncertainty Scale-12 (IUS-12) has proven to be a sound measure of intolerance of uncertainty with excellent validity and reliability. Although research has supported a two-factor structure of IUS-12, most recent studies also suggest a bifactor model. The present study examines the factorial structure, validity, and reliability of the Greek version of IUS-12 with a sample of 959 university students (66.6% female) aged 19.63 years (SD = 3.20). Confirmatory Factor Analysis showed that although the two-factor solution adequately fit the data, the bifactor model better fit with IU total as an underlying one-factor. Internal consistency and validity were excellent for the total IUS-12 and Prospective and Inhibitory Anxiety subscales. Our findings support recent findings concerning the factorial structure of IUS-12 and the scale's psychometric qualities in a Greek undergraduate sample. We expect that future research with clinical samples will confirm the screening and clinical utility of IUS-12.

**Keywords:** intolerance of uncertainty; confirmatory factor; psychometric properties; validation; Greek





## 1. Introduction

In 1993, Krohne [1] proposed that when an individual faces ambiguous stimuli, they experience a state of uncertainty. In studies exploring worry processes, Butler and Matthews found that worriers tend to interpret ambiguous events as threatening [2]. Furthermore, Metzger et al. found that worriers took longer to make decisions in ambiguous situations [3]. Therefore, researchers proposed that it may be uncertainty that led to the perception of threat in ambiguous situations and that the 'not knowing' led to worry. Intolerance of Uncertainty (IU) has been described as the tendency to avoid an uncertain state and exhibit negative responses to uncertain situations on cognitive, emotional, and behavioral levels [4]. Individuals with high IU tend to negatively misinterpret ambiguous situations on a cognitive level [5], while behaviorally, they attempt to avoid such situations or engage in specific behaviors to ensure certainty. IU is typically conceptualized as a higher-order construct composed of two lower-order dimensions and has been characterized as a transdiagnostic cognitive bias [6] that plays a role in developing and maintaining psychopathology [7,8].

Intolerance of Uncertainty was identified by the Obsessive-Compulsive Cognitions Working Group [9] as one of the six belief domains central to Obsessive-Compulsive Disorder (OCD). Relevant as well as consequent research also found a strong relationship between OCD phenomena and IU [10,11] and identified IU as a critical process in Generalized Anxiety Disorder (GAD) and other anxiety disorders [4,12–17]. In their study, McEvoy and Mahoney (2011) found a strong association between intolerance of uncertainty (IU) and anxiety symptoms in their sample of individuals seeking treatment for anxiety.

Specifically, they found that higher levels of IU were significantly associated with greater severity of anxiety symptoms. The authors also found that different aspects of IU were related to different anxiety symptoms. For example, inhibitory IU (worries about making decisions) was found to be most strongly associated with social anxiety symptoms, whereas prospective IU (worries about the future) was most strongly associated with generalized anxiety symptoms [17].

IU has also been associated with Major Depressive Disorder (MDD) [17–19]. Yook et al. (2010) investigated the relationship between intolerance of uncertainty (IU) in individuals with Major Depressive Disorder (MDD) and Generalized Anxiety Disorder (GAD). They recruited 35 participants with MDD, 35 participants with GAD, and 35 healthy control participants. The results showed that both the MDD and GAD groups had significantly higher levels of IU, worry, and rumination than the healthy control group. Additionally, in the MDD group [18], IU seems to lead to over-identification of potential problems or negative problem orientation [4], hypothesized to lead to depressive symptoms [18]. Additionally, people with high IU may adopt a pessimistic certainty for future events, expecting the worst, which also, in turn, may predispose them to depression [18,19]. Saulnier, Allan, Raines, and Schmidt studied the relationship of IU and its two IU lower-order dimensions to depressive symptoms and found that the general IU factor related to cognitive and Affective/Somatic symptoms of depression [20]. Nevertheless, Inhibitory Anxiety IU (the fear of future unpredictable events) is associated with the Cognitive factor of depression but not the Affective/Somatic factor. On the other hand, no relation was found between Prospective Anxiety IU (avoidance due to fear of uncertain events) and the Cognitive or Affective/ Somatic symptoms, pointing thus to a differential way that IU dimensions play in the etiology of depressive symptoms [20].

Various measures have been developed to assess intolerance of uncertainty, such as the Uncertainty Response Scale [21], the Tolerance of Ambiguity Scale [22], or the Intolerance of Uncertainty/Perfectionism subscale of the Obsessive Beliefs Questionnaire [23,24]. One of the most widely used measures is the Intolerance of Uncertainty Scale (IUS) [4]. The IUS was originally developed in French to assess responses to ambiguous situations, uncertainty, and future events [4]. The original IUS consisted of 27 items derived from a larger pool, had excellent internal consistency and good test-retest reliability, and moderate to good convergent validity with measures associated with Depression, Anxiety, and Worry.

Further research in the translated English version indicated strong support for a four-factor structure and maintained excellent internal consistency [25]. Consequent factor analytic studies, although they supported internal consistency and reliability, either confirmed the four-factor solution [17,26] or introduced five- and six-factor solutions [27] or one-factor solutions [28].

Carleton, Norton, and Asmundson (2005) conducted research with different model structures testing four and five IUS model structures [29]. They found that none of the IUS models they tested fit the data appropriately. As a result, they proceeded in the process of item reduction based on Norton's notion [27], which resulted in the IUS-12 item scale. The IUS-12 demonstrated excellent internal consistency and a positive correlation with the original 27-item version. In an independent sample, a confirmatory factor analysis revealed a two-factor structure for the IUS-12. The internal consistency of each factor was deemed acceptable, and the two factors remained moderately correlated. The two factors were named Prospective Anxiety and Inhibitory Anxiety. Prospective anxiety reflects the desire for predictability and active engagement in seeking certainty, while inhibitory anxiety refers to the tendency to experience cognitive and behavioral paralysis in the face of uncertainty.

A consequent study confirmed the factor structure of IUS-12 in an OCD sample. In the study by Jacoby et al. (2013), the researchers aimed to confirm the factor structure of the 12-item version of the Intolerance of Uncertainty Scale (IUS-12) in a sample of patients with obsessive-compulsive disorder (OCD). They conducted a confirmatory factor analysis and found that the two-factor model initially proposed for the IUS-12 provided a good fit to the data. The two factors were labeled as "Prospective IU" (i.e., the desire for predictability and

control) and "Inhibitory IU" (i.e., the tendency to become paralyzed by uncertainty). The researchers concluded that the two-factor model of the IUS-12 was a good representation of intolerance of uncertainty in OCD patients and that the measure may be helpful in assessing and treating this population [30]. IUS-12 also demonstrated high internal consistency and moderate convergent validity. Finally, results showed a specific correlation between the IUS-12 and OCD symptom dimensions [30].

Several studies of translated and adapted versions of IUS-12 also supported the good internal consistency and reliability qualities of IUS-12 in different languages and cultures and strong evidence of a mainly two-factor model and often a bifactor model. IUS-12 has been translated into Dutch [31], Turkish [32], Italian [33,34], Chinese [35,36], Serbian [37], and Brazilian Portuguese [38]. Furthermore, IUS has been adapted for children showing good psychometric properties [39–41].

A systematic review of factor-analytic studies showed that two factors with 12 items emerged throughout the exploratory analyses. Birrell, Meares, Wilkinson, and Freeston (2011) demonstrated the stability of models containing these two factors in subsequent confirmatory studies [42]. However, more recent studies prioritized a bifactor model over a two-factor model [43,44]. The study by Hale et al. (2016) aimed to address previous inconsistencies in the factor structure of the intolerance of Uncertainty Scale-12 and determine the best scoring approach for the measure. The researchers conducted a confirmatory factor analysis and found that the bifactor model provided a superior fit to their undergraduate sample data, indicating that the IUS-12 may be better represented as a unidimensional construct with two specific factors. The researchers suggested that a summed total score of the IUS-12 was a more reliable index of intolerance of uncertainty than the scores of the two subscales separately [43]. Additionally, Lauriola et al. (2018) found that the bi-factor model fitted this student sample data better than alternative models and that this general factor accounted for 80% of the item variance [44]. This underlying factor explains most of the reliable variance across the IUS-12 items and suggests that a total score is an appropriate IU index [45,46]. Furthermore, Wilson et al. (2019) aimed to examine the psychometric properties of the IUS-12 in individuals with Generalized Anxiety Disorder (GAD). The researchers conducted a confirmatory factor analysis and found that the originally proposed two-factor model of the IUS-12 had a poor fit to the data. Findings showed that the bi-factor model demonstrated the best fit. The test had good validity and reliability and showed responsivity to treatment [47].

To the best of our knowledge, there are no studies for the IUS-12 psychometric properties in the Greek language. Therefore, the present study aimed to validate and examine the psychometric properties and the factor structure of IUS-12 in a Greek-speaking non-clinical sample. Part of this was the examination of the IUS-12 convergent and divergent validity. We expected that the IUS-12 would have higher correlations to uncertainty-relevant constructs and lower correlations to less uncertainty-related variables.

## 2. Materials and Methods

### 2.1. Participants

Participants were 959 students from two universities in Thessaloniki with a mean age of 19.63 years (SD = 3.2, R = 18–50). The majority of the participants were female (66.6%). At the end of a weekly undergraduate class, students were invited to participate. Participation was voluntary, and no money or credits were given to participants. The ethics committee of the lead university (University of Macedonia) approved the research.

### 2.2. Measures

Intolerance of Uncertainty Scale -Short form (IUS-12) [29]: IUS-12 is a 12-item self-report measure evaluating one's tendency to find uncertainty as upsetting and distressing. The responses are on a 5-point Likert scale ranging from 1 (not at all characteristic of me) to 5 (entirely characteristic). IUS-12 consists of a 7-item Prospective Anxiety subscale evaluating desire for predictability and cognitive appraisals about future uncertainty and a

5-item. The inhibitory Anxiety subscale assesses one's behavioral inhibition or avoidance when faced with uncertainty. Despite the reported multifactor structures, the IUS is most commonly summed as a total scale score [48]. The IUS-12 has been found to have excellent internal consistency and convergent and discriminant validity. Internal consistency was excellent in the present study (Cronbach's alpha = 0.87).

Obsessive Beliefs Questionnaire- 44 (OBQ-44) [24]: The OBQ-44 evaluates six belief domains linked to OCD, consisting of 44 items split into three subscales: Responsibility and Threat Estimation (Cronbach's a = 0.93), Perfectionism and Intolerance of Uncertainty (Cronbach's a = 0.93), and Importance and Control of Thoughts (Cronbach's a = 0.89). In this study, the OBQ-44 displayed excellent internal consistency with a Cronbach's alpha of 0.90. For this particular study, only the 16-item Perfectionism/Intolerance of Uncertainty subscale was used, which had a Cronbach's alpha of 0.80.

Emotion Regulation Questionnaire (ERQ) [49]: The ERQ is a 10-item self-report scale designed to measure the tendency to regulate emotions by cognitive reappraisal and/or expressive suppression. ERQ does not have an overall score but consists of two independent subscales, (1) Cognitive reappraisal, which is a form of cognitive alteration that involves construing a potentially emotion-eliciting situation in a way that changes its emotional impact, and (2) Expressive Repression, which is a form of response modulation that involves inhibiting ongoing emotion-expressive behavior. The answers are on a Likert scale ranging from 1 (I totally disagree) to 7 (I totally agree), and higher scores indicate greater use of emotion regulation strategies. ERQ has high reliability and validity [46]. Karademas, Tsalikou, & Tallarou used the ERQ with a sample of Greek cardiology patients and found good reliability coefficients for the scale [50]. Cronbach's alphas of the Cognitive reappraisal and the Expressive Repression subscales in the present study were 0.85 and 0.73, respectively.

Depression Anxiety Stress Scale- 21 (DASS- 21) [51]: The DASS-21 evaluates negative emotions and the intensity with which the individual experiences these feelings. It consists of 21 items separated into three subscales: (1) Depression (Cronbach's a = 0.88), (2) Anxiety (Cronbach's a = 0.79), and (3) Stress (Cronbach's a = 0.78). The Greek version of DASS-21 had excellent internal consistency with Cronbach's alpha = 0.91 [52]. For the present study, we used only the DASS-21-Depression subscale. The DASS-21-Depression scale assesses dysphoria, hopelessness, devaluation of life, self-deprecation, lack of interest/involvement, anhedonia, and inertia. In the present study, the DASS-21 Depression subscale had Cronbach's alpha = 0.85.

*2.3. Procedure*

2.3.1. Translation and Cross-Cultural Adaptation

Two independent bilingual researchers translated the IUS-12 from English into Greek, and another bilingual researcher back-translated the Greek version into English. Differences were discussed until the researchers reached a consensus, according to the Beaton, Bombardier, Guillemin, and Ferraz (2000) guidelines [53].

2.3.2. Data Selection

Participants were university undergraduates from the University of Macedonia (81.6%) and the Aristotle University of Thessaloniki (18.4%) and were recruited from university classes. Written informed consent was obtained from all participants after a thorough description of the study. Individuals who agreed to participate were given to complete all four questionnaires in hard copy, and collected data were digitalized later. The data analysis was done by using SPSS 21 and AMOS 21.

**3. Results**

*3.1. Factor Analysis*

Although the determination of the appropriate sample size may be an issue in Structural Equation Modelling (SEM), there is no consensus regarding the acceptable sample size

for SEM. Usually, *N* = 100–150 is considered a minimum acceptable sample size for SEM, while some researchers consider an even larger sample size [54–56]. Another approach is to determine the sample size in light of the number of observed variables. Bentler and Chou suggest that a ratio of 5 cases per variable would be sufficient for normally distributed data [57]. A widely accepted rule of thumb is ten observations per indicator variable in setting a minimum bound of adequate sample size [58]. Considering the above suggestions and having a sample size of 959 in the present study, we randomly extracted from the primary data pool 20% of the participants using SPSS to perform Confirmatory Factor Analysis; this resulted in a sample of 139 participants for the CFA.

Using AMOS 21 software, we conducted a confirmatory factor analysis to assess whether the hypothesized underlying structure of the IUS-12 demonstrated a good fit. An inferred hypothesis about item loadings and factor structure of the IUS-12 was based on previous studies supporting a two-factor solution in both non-clinical [29] and mixed clinical samples [6,17]. We examined one-factor, two-factor, and bifactor solutions using maximum likelihood estimation with the correlation matrix. Model fit was determined using (1) chi-square/df, (2) Comparative Fit Index (CFI; values are permissible > 0.80 and preferably > 0.90 [59]), (3) Goodness of Fit (GFI; acceptable values are > 0.90, and preferably > 0.95 [56], (4) Adjusted Goodness of Fit (AGFI; values should be > 0.80 [56], (5) Root Mean Square Error of Approximation (RMSEA; values are moderate to good < 0.10 [56], (6) Expected Cross Validation Index (ECVI; lower values indicate better fit [60] (7) Standardized Root Mean Square Residual (SRMR; values of 0.08 or lower is often considered an indicator of good model fit) [59].

The one-factor model did not have a very good fit. Most indices are either lower or higher than the recommended cut-off scores indicating that the model is poorly supported. Therefore, we proceeded with the two-factor model.

According to most indices in the current study, the two-factor solution provided a poor to adequate fit to the data; the GFI, AGFI, and RMSEA were below the acceptable cut-off point, but were very close to the acceptable threshold. All items loaded highly on their respective factor, with Standardized Regression Weights ranging from 0.51 to 0.74. As expected, the two factors strongly correlated (r = 0.85). However, the two-factor solution (Figure 1) appeared to be superior to the one-factor model.

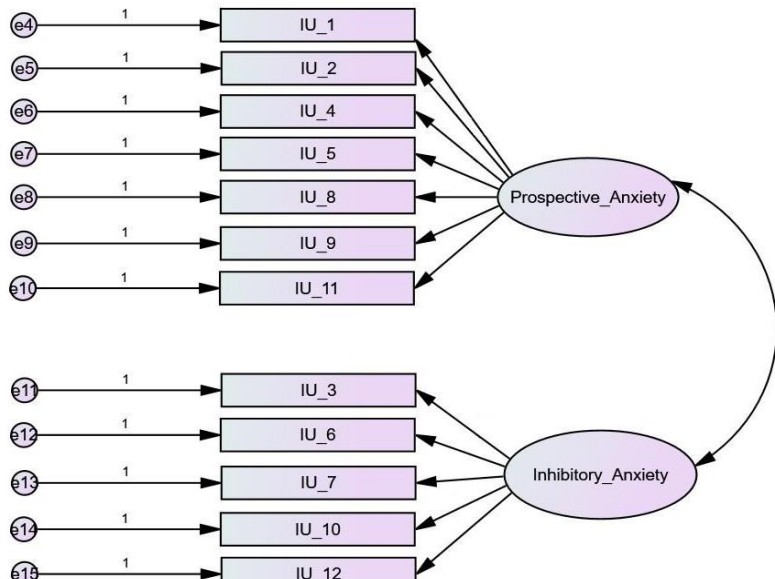

**Figure 1.** Confirmatory factor analysis of the Intolerance of Uncertainty Scale (IUS-12), two-factor structure model.

We further examined the bifactor model (Figure 2), which recent studies suggest that it provides a better fit of data with IU total as an underlying one-factor. The bifactor model

showed a better fit of data and an improvement compared to the two-factor solution. All items loaded highly on underlying IU total factor, with Standardized Regression Weights ranging from 0.56 to 0.68 (Table 1).

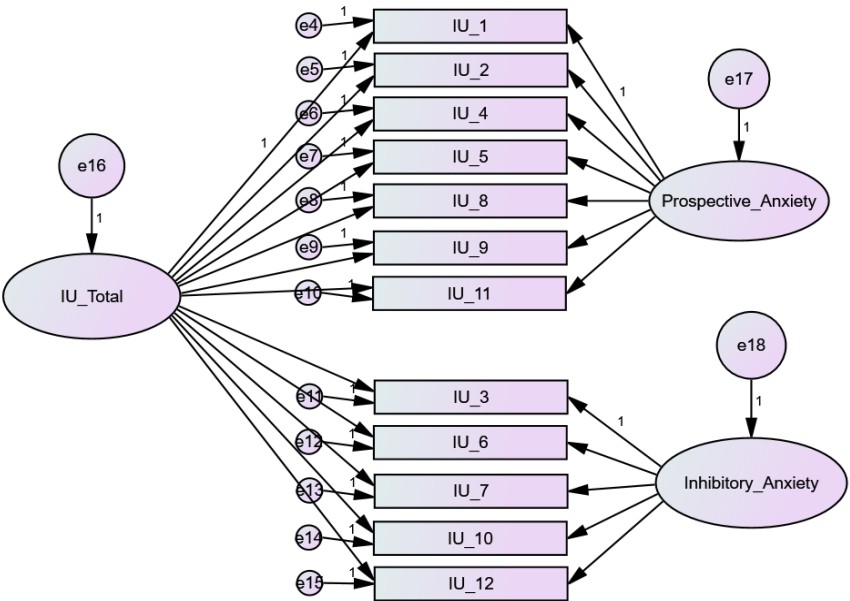

**Figure 2.** Confirmatory factor analysis of the Intolerance of Uncertainty Scale (IUS-12), bi-factor structural model.

**Table 1.** IUS-12 Confirmatory Factor Indices.

|  | $x^2$/df | CFI | GFI | AGFI | RMSEA | SRMR | ECVI |
|---|---|---|---|---|---|---|---|
| 1-Factor | 2.81 | 0.82 | 0.85 | 0.78 | 0.11 | 0.10 | 1.44 |
| 2-Factor | 2.61 | 0.85 | 0.86 | 0.79 | 0.10 | 0.10 | 1.36 |
| Bi-Factor | 1.99 | 0.93 | 0.91 | 0.84 | 0.08 | 0.070 | 1.13 |

CFI: Comparative Fit Index, GFI: Goodness of Fit, AGFI: Adjusted Goodness of Fit, RSMEA: Root Mean Square Error of Approximation, ECVI: Expected Cross Validation Index, SRMR: Standardized Root Mean Square Residual.

### 3.2. Reliability

Table 2 shows mean scores, standard deviations, and internal consistency coefficients (Cronbach's Alpha) for IUS-12 total score and its subscales. Since we wanted to have a raw comparison of our findings with the ones from similar studies, we collected reported descriptive statistics of previous studies with university samples for IUS-12 (Table 3). Greek students seem to show similar total scores with other studies with a similar population.

**Table 2.** Internal Consistency (Cronbach's a) of IUS-12 mean scores and standard deviation.

| IUS-12 | α | M | SD |
|---|---|---|---|
| IUS-12 Total | 0.88 | 30.73 | 8.83 |
| IU Prospective Anxiety subscale | 0.81 | 18.86 | 5.41 |
| IU Inhibitory Anxiety subscale | 0.80 | 11.87 | 4.20 |

**Table 3.** Descriptive statistics of IUS-12 in university student samples.

| Previous Research | Sample | N | Mean | SD |
|---|---|---|---|---|
| Khawaja & Yu (2010) [61]. | Undergraduates/ Australia | 56 | 30.62 | 9.98 |
| Helsen, Van den Bussche, Vlaeyen, & Goubert (2013) [31]. | Undergraduates/ The Netherlands | 967 | 29.41 | 7.56 |
| Fergus, Bardeen, & Wu (2013) [62]. | Undergraduates/ USA | 121 | 26.76 | 9.62 |
| Lauriola, Mosca, & Carleton (2016) [34]. | Undergraduates/ Italy | 609 | 29.69 | 8.06 |
| Shihata, McEvoy, & Mullan (2018) [45]. | Undergraduates/ Australia | 506 | 33.25 | 9.80 |
| Yao, Qian, Jiang, & Elhai (2020) [36]. | Undergraduates/ China student | 696 | 37.21 | 7.02 |
| Huntley, Young, Smith, & Fisher (2020) [63]. | Undergraduates/ UK | 288 463 | 33.23 28.71 | 10.56 9.41 |
| Kretzmann, & Gauer (2020) [38]. | Community/ Brazil | 704 | 38.70 | 10.20 |

### 3.3. Convergent Validity

The convergent validity of IUS-12 was assessed by correlating the IUS-12 total with the Perfectionism/Certainty subscale of the OBQ-44. Table 4 shows bivariate Pearson correlations between the IUS-12 and its subscales and the OBQ-44 Perfectionism/Certainty subscale (Table 4).

**Table 4.** Bivariate Correlations between IUS-12 and OBQ-44 Perfectionism/certainty.

| | Perfectionism/Certainty |
|---|---|
| IUS-12 Total | 0.502 ** |
| IUS-12 Prospective Anxiety | 0.497 ** |
| IUS-12 Inhibitory Anxiety | 0.414 ** |

** Correlation is significant at the 0.01 level (2-tailed).

### 3.4. Divergent Validity

The divergent validity of IUS-12 was assessed by correlating the IUS-12 total and its subscales with the DASS-21 Depression subscale and the Emotion Regulation Questionnaire subscales (Table 5).

**Table 5.** Bivariate Correlations between IUS-12 and the DASS-21 Depression subscale Emotion Regulation Questionnaire.

| | Depression Anxiety Stress Scale | Emotion Regulation Questionnaire | |
|---|---|---|---|
| IUS-12 | Depression | Cognitive Reappraisal | Expressive Suppression |
| IUS-12 Total | 0.478 ** | −125 ** | 308 ** |
| IU Prospective Anxiety | 0.387 ** | −0.091 ** | 273 ** |
| IU Inhibitory Anxiety | 0.505 ** | −145 ** | 294 ** |

** significance at the 0.01 level (2-tailed).

### 3.5. Gender Differences

There were no significant variations between genders in the total score (female M = 30.85, SD = 8.83; male M = 30.68, SD = 8.61), t(954) = −0.277, $p > 0.05$, as well as

the Inhibitory Anxiety IU subscale (female M = 12.04, SD = 4.17; male M = 11.51, SD = 4.15), t(954) = 0.974, $p$ > 0.05 and the Prospective Anxiety IU subscale (female M = 18.82, SD = 5.40; male M = 19.17, SD = 5.24), t(954) = −1.835, $p$ > 0.05 There is no reason to assume substantive sex-based differences on the IUS-12

## 4. Discussion

Intolerance of uncertainty has emerged as a key construct related to the development and maintenance of various disorders, especially GAD and OCD [10,30]. The Intolerance of Uncertainty Scale, either the 27-item or the 12-item version, is the most widely used assessment tool for intolerance of uncertainty. The objective of this study was to evaluate the psychometric features of the IUS-12 questionnaire in a non-clinical Greek-speaking population.

The findings indicate that the Greek version of the IUS-12 maintains the same psychometric properties as the original version. The internal consistency of both the overall scale and the two subscales was excellent, as previously reported in other studies [31,42,61]. Convergent validity was evaluated by analyzing the correlation between the IUS-12 scale and the 'Perfectionism/Certainty' subscale of the OBQ-44, which measures a similar construct. The high correlation observed confirms that the IUS-12 has good convergent validity [31,62].

Divergent validity was examined by assessing the correlation between the IUS-12 and the DASS-21 'Depression' subscale and the Emotion Regulation Scale's two subscales. Since intolerance of uncertainty is conceptualized as 'cognitive, emotional and behavioral reactions to uncertainty in everyday life situations' [4], IU would likely correlate with measures of depression, something that was confirmed in the Gentes & Ruscio meta-analysis [10]. Nevertheless, the association of IUS-12 to Depression was less pronounced than the association between IUS-12 and the OBQ-44 'Perfectionism/Certainty subscale. This lower IUS-12 and Depression association confirms other studies' findings [25,30,31]. Interestingly the Inhibitory Anxiety IU subscale showed a stronger correlation with the depression subscale, while the Prospective Anxiety IU subscale showed a lower correlation; this finding partly supports previous findings showing that Inhibitory Anxiety IU correlates uniquely with cognitive symptoms of depression, while Prospective Anxiety IU shows a moderate correlation with cognitive and affective symptoms of depression [20]. In addition, we evaluated the divergent validity of the IUS-12 by comparing it with another measure of a different construct, Emotion Regulation (ERQ), instead of a symptom scale, which is typically unrelated to intolerance of uncertainty. Our results revealed a weak negative correlation between IUS-12 and the ERQ Cognitive Reappraisal subscale and a moderate-to-low positive correlation between IU and the ERQ Expressive Repression subscale. Both correlations were lower than those between the IUS-12 and the 'Depression' and 'Perfectionism/Certainty' scales. These findings are consistent with our hypothesis that IUS-12 has good construct validity. These findings suggest that the IUS-12 measures intolerance of uncertainty as a process construct rather than a symptom and can be used to distinguish it from other constructs, such as emotion regulation.

The confirmatory factor analysis performed in this study revealed a poor fit of the model, which did not exactly replicate the two-factor structure observed in other studies [29,42,64]. Nevertheless, the two-factor model was still superior to the one-factor model and approached the adequate threshold based on specific indices, suggesting that it may still provide a reasonable approximation of the underlying structure of the IUS-12. Furthermore, the bi-factor model showed a better fit, consistent with more recent studies [43,44,47], indicating that the general intolerance of uncertainty factor is a reliable construct, and the IUS-12 can be represented as a unidimensional model. This finding suggests that the total score of intolerance of uncertainty may be a more reliable index than the scores of the two subscales separately when assessing intolerance of uncertainty.

It is important to acknowledge several key limitations of this study. First, the participants were all university students, which may limit the generalizability of the results to the broader Greek population. Additionally, the study was conducted in a non-clinical sample, which means that further research with clinical populations is necessary to assess the psy-

chometric properties of the IUS-12 and its ability to differentiate pathological intolerance of uncertainty. Further studies are needed to evaluate the reliability and validity of the IUS-12 in different contexts and populations.

Although there were some limitations to the current study, the results support the good psychometric properties of the IUS-12 in a Greek non-clinical sample. Given the scarcity of research on the IUS-12 in a Greek clinical population, future studies may benefit from using diverse samples and methodological approaches to investigate the stability of the bifactor structure over time.

The findings of the present study may inform the selection of the IUS-12 as a reliable and valid measure of intolerance of uncertainty in research and clinical practice. Nonetheless, it is important to note that further research is required to confirm and expand on these findings in different populations and contexts. The IUS-12 may be useful for assessing intolerance of uncertainty in various clinical settings, such as anxiety and obsessive-compulsive disorders. Additionally, future studies may investigate the relationship between intolerance of uncertainty and other constructs and cognitive biases to further our understanding of this construct. Overall, the present study contributes to the growing body of research on the IUS-12 and its factor structure and highlights the need for further research in this area.

## 5. Conclusions

The results of this study offer compelling evidence for the practicality and psychometric strength of the Greek version of the IUS-12 questionnaire as a tool for measuring intolerance of uncertainty. Given its shorter length, the IUS-12 questionnaire could encourage more extensive assessments of intolerance of uncertainty in both clinical and non-clinical populations.

**Author Contributions:** Conceptualization, G.S. and A.N.; methodology, G.S. and A.N.; formal analysis, G.S. and A.N.; data curation, G.S. and A.N.; writing—original draft preparation, A.N.; writing—review and editing, G.S.; supervision, G.S. All authors have read and agreed to the published version of the manuscript.

**Funding:** This research received no external funding.

**Institutional Review Board Statement:** The study was conducted in accordance with the Declaration of Helsinki and approved by the Institutional Review Board (or Ethics Committee) of University of Macedonia (13/4-6-2018).

**Informed Consent Statement:** Informed consent was obtained from all subjects involved in the study.

**Data Availability Statement:** The data presented in this study are available on request from the corresponding author. The data are not publicly available because this study is part of ongoing research.

**Conflicts of Interest:** The authors declare no conflict of interest.

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
