# Peer review of "Factor Structure, Validity and Reliability of the Intolerance of Uncertainty Scale -12 (IUS-12) in a Greek Undergraduate Sample"

_2673-5318, doi:10.3390/psychiatryint4020010_

Round 1

Reviewer 1 Report

This is a high quality and well-presented paper. The intolerance of uncertainty scale is an important scale and considering its nuances in language, validation in other languages is both useful and of relevance. The authors have used a scientifically sound method to validate the scale in Greek and they have appropriately discussed the limitations of their study. The conclusions are sound.

Author Response

Thank you so much for your review and your kind response.

Reviewer 2 Report

This is an interesting piece and I always appreciate authors who engaged with validation studies although they are rarely published in scientific journals. I have no particular comments to improve the paper as it is okay as it is. INdeed, a validation study is very formal and structured and what matters is the result of study. So, in my review, I will only focus on the results of the paper.

My concerns are the followings: 

First, I would like to ask you to provide the rationale for the indices of fits that you have chosen. It is quite uncommon to use AGFI, can you provide an explanation of why you have chosen all these indices? Also, I wonder why you did not use SRMR, why did you not include this index?

Second, I really appreciate that you reported the scan of AMOS so readers can see what has been done actually. However, if you report beta indices in Figure 1, you should also report them in Figure 2. Likewise, I wonder whether the Figure reports standardized or unstandardized betas, can you provide information about it?

The rest of the paper is okay.

Author Response

Thank you for your review and your very valid observations. You have raised some interesting points and we would like to address each point.

  1. We chose to present AGFIs in terms of comparing AGFI values between different models, a higher AGFI value for a proposed model relative to a baseline model (i.e., a simpler model with fewer parameters) suggests that the proposed model provides a better fit to the data than the baseline model. Hu and Bentler (1999) discuss the use of incremental fit indices in general, and the use of AGFI in particular, for evaluating model fit in structural equation modelling. They suggest that a higher AGFI value for a proposed model relative to a baseline model suggests that the proposed model provides a better fit to the data than the baseline model. Our rationale was to show the relative better fit of a bifactor model rather than the other two models. Byrne (2010) and Kline (2011) also addressed the use of incremental fit indices, such as AGFI, and the comparison of these indices between different models to assess the improvement in fit of a more complex model relative to a simpler model.
  2. As for the SRMR, you did an excellent point about including that index and after careful consideration of the bibliography we decided that it would be an oversight to not include it. Therefore, we made the appropriate adjustments in our table in order to show SRMR.
  3. Finally, to answer your question in the Amos it shows the unstandardized weights. However, we decided to include these scans in order to show visually the difference of a two-factor vs a bifactor model, that is why we did not include the one factor model. Thus, we decided to include a scan without the weights as both models have both standardized and unstandardized weights above .50 which we report in text.

Round 2

Reviewer 2 Report

Great job, good luck with the publication